# The Species Effect: Differential Sphingosine-1-Phosphate Responses in the Bone in Human Versus Mouse

**DOI:** 10.3390/ijms25105118

**Published:** 2024-05-08

**Authors:** Kathryn Frost, Jonathan W. Lewis, Simon W. Jones, James R. Edwards, Amy J. Naylor, Helen M. McGettrick

**Affiliations:** 1Institute of Inflammation and Ageing, University of Birmingham, Birmingham B15 2TT, UK; kxf143@student.bham.ac.uk (K.F.); j.lewis.2@bham.ac.uk (J.W.L.); s.w.jones@bham.ac.uk (S.W.J.); a.naylor@bham.ac.uk (A.J.N.); 2Botnar Research Centre, University of Oxford, Oxford OX3 7LD, UK; james.edwards@ndorms.ox.ac.uk

**Keywords:** osteoblast, osteoclast, sphingosine-1-phosphate (S1P), human, mouse, bone

## Abstract

The deterioration of osteoblast-led bone formation and the upregulation of osteoclast-regulated bone resorption are the primary causes of bone diseases, including osteoporosis. Numerous circulating factors play a role in bone homeostasis by regulating osteoblast and osteoclast activity, including the sphingolipid—sphingosine-1-phosphate (S1P). However, to date no comprehensive studies have investigated the impact of S1P activity on human and murine osteoblasts and osteoclasts. We observed species-specific responses to S1P in both osteoblasts and osteoclasts, where S1P stimulated human osteoblast mineralisation and reduced human pre-osteoclast differentiation and mineral resorption, thereby favouring bone formation. The opposite was true for murine osteoblasts and osteoclasts, resulting in more mineral resorption and less mineral deposition. Species-specific differences in osteoblast responses to S1P were potentially explained by differential expression of S1P receptor 1. By contrast, human and murine osteoclasts expressed comparable levels of S1P receptors but showed differential expression patterns of the two sphingosine kinase enzymes responsible for S1P production. Ultimately, we reveal that murine models may not accurately represent how human bone cells will respond to S1P, and thus are not a suitable model for exploring S1P physiology or potential therapeutic agents.

## 1. Introduction

Throughout life, the skeleton is continuously remodelled, adapting to mechanical stress and repairing accrued damage [1]. Bone remodelling is an intricate, tightly regulated process that is accomplished by two major bone cell types: osteoblasts are responsible for forming new bone, whilst osteoclasts are tasked with resorbing and removing old or damaged bone tissue [2]. Sphingosine-1-phosphate (S1P) is a known regulator of bone homeostasis and has been implicated in driving the onset and/or progression of numerous bone diseases (e.g., as reviewed by [3]). Despite this, there remains conflicting evidence surrounding the functional responsiveness of osteoblasts and osteoclasts to S1P, making interpretation of the available data and translation of S1P related therapies in the bone disease arena fraught with difficulty.

Osteoblasts differentiate from mesenchymal lineage progenitors, whilst osteoclasts are derived from the fusion of monocytes, which are haematopoietic in origin [2]. Whilst several studies have investigated the functional consequence of S1P on osteoblastogenesis, inconsistencies in the reported expression pattern of the five S1P receptors on osteoblasts and their precursors [4,5], and how these change upon maturation [6,7], adds to the complexity within the field. Consequently, S1P has been reported to drive osteoblast maturation and/or mineralisation in murine (C3H10T1/2 [4], MC3T3-1 [8]) and human (SaOS-2 osteosarcoma [9]) cell lines. Conversely, S1P inhibited the mineralisation of murine primary osteoblast progenitors (derived from dental pulp) in culture [10]. Only two studies have investigated the role of S1P in osteoclastogenesis and both used murine cells [11,12]. S1P negatively regulated osteoclastogenesis in cells cultured from murine whole bone marrow (BM) aspirates—the increase in intracellular S1P reduced p38 signalling and limited the expression of the osteoclast maturation marker, tartrate resistant acid phosphatase (TRAP) [11]. In contrast, extracellular S1P signalling through S1P receptor 2 (S1PR2) drove formation of TRAP-positive osteoclasts from murine BM-derived monocytes (BMDM) from ICR mice [12]. Of note, this response was not seen when treating murine BM macrophage-like cells from C57BL/6 mice with exogenous S1P in culture [11]. Clearly, further research is required to ascertain the exact impact of S1P on bone remodelling, specifically within the human setting.

The utilization of murine cells for studying human conditions has been a topic of extensive debate within the scientific community and is particularly relevant in the osteology field, where species (mouse vs. human) differences have been reported in the response of osteoblasts and osteoclasts in vitro and in vivo [13,14,15]. In this study, we aimed to compare the effects of S1P stimulation on human and murine osteoblasts and osteoclasts to ascertain whether species specific responses existed that could account for the conflicting reports. Our data revealed disparate responses and suggested variations in receptor expression and metabolism between species. These findings demonstrate the care that must be taken when interpretating in vitro data and seeking to translate findings into the clinic.

## 2. Results

### 2.1. S1P Has Differential Effects on Osteoblastogenesis in Mice and Humans

Initially, we investigated the functional impact of a single concentration of S1P on osteoblast maturation using primary murine and human osteoblasts, two murine cell lines (ST2 and MC3T3-E1), and a human cell line (hFOB) (Figure 1A–E). S1P treatment significantly reduced osteoblast maturation, as measured by ALP activity, in all murine cells used compared to untreated controls (Figure 1A–C). By contrast, ALP activity was significantly increased in primary human osteoblasts and hFOB treated with S1P compared to untreated cells (Figure 1D,E). These data indicate species-specific osteoblast responses to S1P.

We hypothesised that this differential response could be caused by differences in the ability of cells to respond to S1P, based on which combination of S1PR are expressed and when. To dissect receptor expression in each species, we analysed publicly available scRNA-seq datasets for expression of the S1P receptors to understand how this pattern differs by osteoblast maturation stage [16,17]. Osteoblasts were identified as *Alpl*, *Bglap*, *Bglap2*, *Omd*, *Runx2*, and *Sp7* positive cells within murine bone marrow isolates, and further subdivided into osteoprogenitors, pre-osteoblasts, and mature osteoblasts for murine cells and mesenchymal stem cells and osteoblasts in the human population (Appendix A). Murine osteoprogenitors had a high expression of *S1pr1* and *S1pr3*, which was lost/diminished upon differentiation into mature osteoblasts (Figure 1F). Whilst the expression pattern for human *S1PR3* was comparable with murine cells, *S1PR1* was conversely increased during human osteoblast differentiation (Figure 1G). No S1P receptor 4 or 5 mRNA was detected in either murine or human precursors or mature osteoblasts, and only low levels of S1P receptor 2 transcript was detected in either murine (Figure 1F) or human (Figure 1G) cells. We subsequently validated these observations using qPCR for *S1PR1-4* in primary murine and human osteoblasts (Figure 1H–M). In contrast to the scRNA-seq data, we observed no effect of osteoblast differentiation on expression of *S1pr1* in murine calvarial osteoblasts (Figure 1H), and a significant increase in *S1PR1* in human osteoblasts (Figure 1K). As with *S1pr1*, we see no changes in *S1pr3* in murine cells upon osteoblast differentiation (Figure 1I), whereas we see a tendency for higher expression of *S1PR3* within mature human osteoblasts (Figure 1L). Unlike the scRNA-seq dataset, we were able to detect S1PR4 expression in osteoblasts and observed a significant reduction or increase in expression following differentiation of murine (Figure 1J) or human (Figure 1M) osteoblasts, respectively. These data suggest that species-specific differences in S1P bioactivity during osteoblastogenesis are due to the differential expression pattern of S1PR on the cells during differentiation and therefore the downstream signalling that is capable of being induced.

Whilst we used a single concentration of S1P in our functional studies above, it is possible that local S1P levels differ between species, due to the intrinsic capability of osteoblasts to produce and/or secrete S1P. To reveal this, we assessed the expression levels of sphingosine kinases (SPHK1 and SPHK2), the enzymes responsible for the phosphorylation of sphingosine to sphingosine-1-phosphate. Analysis of the osteoblast maturation datasets revealed that both murine and human osteoblasts express *Sphk1* and to a lesser extent *Sphk2*, with a tendency for increased *Sphk1* expression during differentiation (Figure 2A,B). By contrast, qPCR revealed no differences in the expression of either kinase or the S1P transporter (*Spsn2*) upon differentiation of murine osteoblasts (Figure 2C–E). However, human osteoblasts showed an upregulation of *SPHK1* and downregulation of *SPNS2* during maturation (Figure 2F–H). Whilst our data clearly show the ability of both murine and human osteoblasts to generate and release S1P to a similar degree, this is unlikely to account for the differential functional response we observe.

### 2.2. Murine Osteoblasts Develop an Inflammatory Phenotype following S1P Treatment

To further explore the response of murine osteoblasts to S1P, primary calvarial osteoblasts treated for 8 days with or without S1P were analysed by bulk RNA-seq. As expected, S1P induced significant transcriptional changes in osteoblasts when compared to the untreated cells (Figure 3A,B). There was a significant down-regulation in the expression of genes associated with osteoblast maturation (*Bglap* and *Bglap2*, Appendix A) in S1P-treated osteoblasts compared to the untreated group, aligning with the decreased ALP activity seen in murine cells (Figure 1A–C). Indeed, pathway analysis revealed down-regulation of genes associated with ossification, biomineralisation, and organ morphogenesis upon S1P treatment of murine pre-osteoblasts (Figure 3C,D). In fact, prolonged exposure to S1P used in this experiment (8 days) induced an inflammatory phenotype within the pre-osteoblasts, resulting in a statistically significant upregulation in several pro-inflammatory cytokines (including *il6*, *tnfrsf9*) and chemokines (e.g., *Cxcl9*, *Cxcl10*, *Ccl11*, Appendix A). Moreover, upregulated genes were related to immune responses, regulation of cytokine production, and blood vessel morphogenesis (Figure 3E,F). These data indicate that in mice S1P alters the bone microenvironment, favouring immune responses and angiogenesis/vascularisation over homeostatic processes of tissue remodelling. To explore if similar immune-related responses occurred in human osteoblasts treated with S1P, we analysed the gene expression of some immune markers upregulated in the murine cells. In contrast to murine osteoblasts, expression of the proinflammatory genes *IL6* and *CXCL9* were significantly decreased by treatment with S1P (Figure 3G,H), whilst *CXCL10* was significantly upregulated (Figure 3I). Collectively, these data suggest species-specific S1P responses linked with induction (mouse) or suppression (human) of pro-inflammatory linked pathways.

### 2.3. S1P Has Differential Effects on Osteoclastogenesis in Mice and Humans

Subsequently, we assessed whether S1P also differentially regulated osteoclast differentiation and resorption activity using primary murine and human osteoclast precursors (monocytes) (Figure 4 and Figure 5). In human cells, S1P treatment significantly reduced the number of TRAP-positive osteoclasts and amount of resorption of dentine disks in response to RANKL-induced osteoclastogenesis when compared to RANKL-only treated cells (Figure 4A–D). Agreeing with these findings, S1P significantly decreased the expression of markers associated with osteoclast differentiation (*ACP5*, *CTSK*), whilst increasing the expression of the macrophage marker *MSR1* (Figure 4E–G), skewing human monocyte differentiation towards a macrophage, rather than osteoclast, phenotype. In contrast, the number of TRAP-positive murine osteoclasts and their resorption on hydroxyapatite coated plates was significantly increased in response to S1P (Figure 5A–D). However, findings were not mirrored at the transcript level for markers of osteoclastogenesis, with no changes in expression of *Acp5*, *Ctsk*, or *Msr1* seen following S1P treatment when compared to non-S1P-treated osteoclasts (Figure 5E–G). As seen with osteoblasts, these data indicate species-specific differences in the role of S1P during osteoclastogenesis. More importantly, S1P plays an inherent role in osteoblast–osteoclast coupling—favouring bone resorption in mice, but bone mineralisation/formation in humans.

### 2.4. Osteoclastogenesis Regulation of S1P Receptor Expression Is Conversed across Species

Given that we observed differential expression of S1P following osteoblast maturation and between mouse and human cells, we analysed expression of S1P receptor transcripts in publicly available scRNAseq datasets of murine bone marrow treated with RANKL (GSE147174) and human bone tissue (GSE162454). Murine and human cells were clustered based on key markers of monocytes (*S100a8*^high^ *S100a9*^high^), macrophages (*Cd16*^high^ *F4/80*^high^), pre-osteoclasts (*Acp5^med^ Atp6c0d2^med^ Ctsk^med^ Mmp9^med^*), or osteoclasts (*Acp5^high^ Atp6c0d2^high^ Ctsk^high^ Mmp9^high^*; Appendix A). Relative expression of the S1P receptor genes (*S1pr1-4*) within each of these clusters was determined. Murine monocytes expressed high levels of *S1pr4*, with little or no expression of *S1pr1*, *S1pr2*, or *S1pr3* (Figure 6A). Comparatively, human monocytes also showed high expression of *S1PR4*, but also expressed elevated levels of *S1PR3* and low levels of *S1PR2* > *S1PR1* (Figure 6B). Differentiation of murine or human monocytes into macrophages or osteoclasts resulted in a reduction in the expression of *S1PR4* transcript, with a concomitant increase in expression of *S1PR2*–albeit levels of this S1P receptor were highest in macrophages compared to osteoclasts (Figure 6A,B). In agreement with the literature, neither murine or human osteoclast precursors or mature osteoclasts expressed any detectable levels of S1P receptor 5 transcripts (Figure 6A,B). These data are supported by qPCR analysis of human PBMC-derived monocytes and murine BMDMs, where we see no difference in S1P receptor 1 gene expression over differentiation in either species (Figure 6C,E), but a decrease in S1P receptor 4 gene expression with differentiation of osteoclasts (Figure 6F,H)—although only the human data were statistically significant. Here we failed to detect *S1pr3* in murine monocytes/osteoclast and very low *S1PR3* expression in humans (Figure 6D,G). Together, these data suggest that S1P receptor expression evolves during osteoclastogenesis, and this pattern of expression is largely conserved between species.

### 2.5. Differential Expression of Sphingosine Kinase Enzymes in Pre-Osteoclast across Species

As there were relatively few species differences in the relative expression levels of the S1P receptors in murine and human pre- and mature osteoclasts that could explain the differential effects of S1P on osteoclastogenesis, we explored the ability of these cells to generate and release S1P, altering both intracellular and extracellular concentration levels. Murine monocytes and macrophages expressed detectable *Sphk2*, but little if any transcripts for *Sphk1* or *Spsn2* (Figure 7A). All three genes were increased in expression upon differentiation towards osteoclasts, however, *Sphk2* expression was consistently higher than *Sphk1* (Figure 7A). Conversely, human monocytes and macrophages expressed detectable amount of *SPHK1*, but little if any transcripts for *SPHK2* or *SPSN2*, whilst *SPHK1* and *SPSN2* levels were increased in osteoclasts, little if any *SPHK2* was detected (Figure 7B). These data suggest that murine osteoclasts use SPHK2, whilst human osteoclasts use SPHK1 to produce S1P. These data were validated using qPCR showing that both enzymes (*Sphk1*, *Sphk2*) saw a trend increase in murine osteoclasts following differentiation, whereas with human osteoclasts only a significant increase in *SPHK1* was seen (Figure 7C–E,G,H). To confirm protein expression of sphingosine kinases, human and murine monocyte and osteoclast lysates were analysed by western blot. Matching gene expression, murine monocytes and osteoclasts had no detectable mSPHK1 protein expression (Figure 7E), whereas mSPHK2 was expressed at higher levels in murine osteoclasts than monocytes, albeit these differences were not statistically significant (Figure 7F). Furthermore, protein expression profiles of the two kinases in human osteoclasts precursors and mature cells also mirrored the gene expression profile, with little if any detectable hSPHK2 protein, whereas hSPHK1 was present in both monocytes and mature osteoclasts—remaining unchanged by differentiation (Figure 7I,J). The differential expression and use of S1P kinase in human and murine osteoclasts may relate to the differences in S1P responses between species.

It is well known that both intracellular and extracellular S1P can have differing impacts on cell activity [3]. Therefore, we explored the impact of extracellular S1P on the protein expression of sphingosine kinases in osteoclasts, specifically focusing on those where we observed detectable protein at baseline. In murine cells, S1P treatment did not alter mSPHK2 expression (Figure 7K). By contrast, S1P tended to decrease hSPHK1 expression, although this was not statistically significant (Figure 7L), but could potentially indicate a change in intracellular S1P within the human osteoclasts.

## 3. Discussion

S1P has been reported to contribute to the migration of bone progenitor cells, bone homeostasis, and cross-talk between osteoblasts and osteoclasts to regulate bone remodelling [3]. We report for the first time that S1P signalling within bone cells shows species differences—whereby S1P is catabolic in murine cells, reducing osteoblastogenesis and increasing osteoclast activity. Conversely, in human cells S1P acts as a pro-anabolic molecule driving bone mineralisation at the expense of bone resorption. These species-specific differences are likely due to significant differences in S1P receptors in osteoblasts and S1P kinases within osteoclasts between mice and humans. Crucially these data highlight the importance of carefully characterising the functional responses of your chosen cell/model to ensure that appropriate conclusions are drawn. In the case of S1P, it is clearly not possible to substitute murine and human cells and would be inappropriate to translating observations from murine models into the human setting when assessing therapeutic efficacy of candidate molecules.

Our main observations agree with studies reporting that S1P acts as a pro-anabolic agent in human osteoblastogenesis (SaOS-2 osteosarcoma [9]) and in a catabolic fashion in murine osteoblasts (derived from dental pulp) in culture [10]. Our findings contrast the published literature using murine cell lines (C3H10T1/2 [4], MC3T3-1 [8]), which report anabolic activity of S1P. Of note, this discrepancy is unlikely to be due to analysis of primary vs. cell lines, as we have matching data for primary calvarial osteoblasts as well as the MC3T3-1 cell line. One possible explanation could be the fact that Higashi et al. used a slightly higher concentration of S1P (2 µM vs. 1 µM) and also included hydrocortisone within their culture media. It is commonly noted that osteoblasts exhibit high expression of S1PR1-3 [18], however, others have suggested that S1PR4 is also present [5]. Indeed, RNAseq datasets revealed a reduction in expression of both S1P receptors 1 and 3 in both human and murine osteoblasts as they mature. By contrast, we reveal a species-specific increase in S1PR1 in human osteoblastogenesis, which is not seen in murine cells—implying it plays a critical role in mediating S1P signalling in mature human osteoblasts. Meanwhile, the S1P receptor 1 agonist (SEW2871) has been reported to enhance proliferation and alkaline phosphatase activity in human osteoblast cell line (h.Ob), suggesting S1PR1 is involved in human osteoblast maturation [19]. The expression pattern for S1P receptor 4 during osteoblast differentiation contrasts between the species, with it being highest in murine osteoblast precursors and human mature osteoblasts. Yet, no other studies have investigated this receptor in osteoblastogenesis. These variations in S1PR expression throughout osteoblast maturation could contribute to the differential responses elicited by osteoblasts following exogenous S1P administration. Further work is now needed to understand the downstream signalling events mediated by the different S1P receptors following binding of S1P and how these link with stemness or maturation of osteoblasts and the species differences in these responses. Little is known about the role the S1P kinases play in regulating osteoblast function, with a single study suggesting inhibition of both kinases blocks osteoblast differentiation [19]. These studies are vital for furthering our understanding of S1P responses in osteoblasts and any potential therapeutics that might target its potential pro-anabolic actions in human cells.

The field of S1P in osteoclastogenesis is in its infancy—with only two previous publications. Our observation that S1P has catabolic actions in murine osteoclasts derived from C57BL/6 mice agrees with an earlier study demonstrating increased differentiation of BMDM from ICR mice into TRAP-positive osteoclasts in vitro [12]. By contrast, Ryu et al. observed no effect of exogenous S1P or addition of the S1PR antagonist (FTY720) on the differentiation of BM macrophages from C57BL/6 in culture [11]. Whilst this discrepancy is unlikely to be mouse strain-specific given that both ourselves and Hsu et al. have used cells from C57BL/6, the differences could be due to the fact that Ryu et al. pretreated their monocytes with MCSF for a much longer period than used in this study and so the starting pre-cursor cells may exhibit more differentiated macrophages than we used. Moreover, intracellular S1P signalling has been reported to negatively regulated osteoclastogenesis of murine aspirates [11]. To date, ours is the only study to investigate the role of S1P in human osteoclastogenesis, reporting the inverse effects of that seen with murine cells (i.e., less osteoclast differentiation and activity). Additional studies are now required to reproduce these data and build confidence that within the human system S1P limits osteoclastogenesis.

We and others show that the expression of S1PR changes during differentiation of primary murine [4,11] and human cells, however, such an effect was not seen with osteoclast cell lines (e.g., RAW264.7 [11]). Of note, in our hands the most notable change was the loss of S1PR4 during osteoclastogenesis, however, this receptor is rarely mentioned within the literature and as such its function in the bone remains unclear. Others have reported that S1PR2 was the most prominent receptor required to support osteoclastogenesis, as evidenced by the ability of the S1PR2 antagonist (JTE013) to effectively impede the formation of TRAP-positive osteoclasts and prevent the formation of activity/bone resorption pits [12]. Hsu et al. further elucidated that S1PR2 governs osteoclastogenesis by regulating podosome-adhesive proteins necessary for monocyte fusion, thereby forming multinucleated osteoclasts [12]. Knockdown of S1PR2 resulted in decreased levels of phosphorylation of several protein kinases (p-PI3K, p-SRC, or p-PYK2) that are pivotal for monocyte adhesion and fusion [12]. We also observe an increase in the amount of S1P receptor 2 transcripts in mature osteoclasts, in both murine and human cells, adding further weight to its potential ability to influence osteoclast function. That said, it is important to note that the changes in S1P receptors occur independent of species and thus it is improbable that they are responsible for the different functional responses seen in murine and human osteoclasts.

The importance of intracellular S1P signalling has already been noted in the context of RANKL-induced osteoclastogenesis in murine BM aspirates, where there is an enhanced gene expression and the activity of SPHK1 and SPHK2, resulting in an increase in both intracellular and extracellular S1P levels [11]. SPHK1 has previously been named as a negative regulator of osteoclastogenesis [11], with siRNA knockdown of SPHK1 in murine BM-derived macrophages leading to an increased number of TRAP-positive osteoclasts. Again, we are the first to examine the expression profile of these kinases in human pre- and mature osteoclasts, reporting a higher expression of SPHK1 transcripts in mature human osteoclasts. The inherent human preference for SPHK1 may indicate an internal control pathway, whereby S1P negatively regulates osteoclastogenesis. Much less is known about the function of SPHK2, with evidence suggesting it can regulate expression of cell membrane organelles, particularly the mitochondria and nuclei [20], as well as inhibiting HDAC1/2 to facilitate epigenetic regulation of gene expression [21]. Yet the role of SPHK2 in gene regulation in osteoclasts has yet to be fully explored. In the murine macrophage cell line RAW264, inhibition of SPHK2 led to a decrease in c-Fos expression and subsequent osteoclastogenesis [22], highlighting the crucial role of intracellular SPHK2 signalling in the preliminary stages of osteoclast formation. Indeed, our data would support this with the increased expression of *Sphk2* in murine osteoclasts upon differentiation. Surprisingly, Sphk2^−/−^ osteoclasts exhibited normal resorptive activity in vitro, despite global Sphk2^−/−^ mice showing reduced trabecular bone mass [23]. It could be argued that the in vivo phenotype is due to the importance of Sphk2 osteoblast function. Indeed, further investigation revealed a decrease in collagen 1 gene expression and a significant impairment of the anabolic response to PTH in these mice, indicating a deficiency in osteoblast function [23] rather than direct effects on osteoclasts. Of note, the balance in enzymatic activity between both kinases critically influences the intracellular concentration of S1P a cell is subjected to, and the potential cargo that could be released into the extracellular microenvironment. Further work is required to establish whether S1P production differs at a species level.

In summary, we demonstrate that S1P is catabolic in murine cells, resulting in increased osteoclast activity, whilst it is pro-anabolic in human cells driving bone mineralisation. Furthermore, we reveal the importance of undertaking detailed investigation into ligand and receptor expression on cells at all relevant differentiation stages and from all relevant species to enable the appropriate interpretation of in vitro signalling pathway data. Whilst differences between species are not rare or unexpected, our data clearly demonstrate that particular care should be taken whilst investigating the role of S1P in the bone remodelling process. Ultimately, we reveal that murine models may not accurately represent how human bone cells will respond to S1P, and thus are not a suitable model for exploring S1P physiology or potential therapeutic agents.

## 4. Materials and Methods

### 4.1. Isolation of Murine Calvarial Osteoblasts

Mice were purchased from Charles River and were maintained in a specific pathogen-free facility, with free access to food and water. Animal studies were regulated by the Animals (Scientific Procedures) Act 1986 of the United Kingdom and performed under Personal Project Licence (PE5985209) at the Biomedical Services Unit, University of Birmingham, which holds a Section 2C Establishment Licence. Approval was granted by the University of Birmingham’s Animal Welfare and Ethical Review Body and all ethical guidelines were adhered to whilst carrying out this study.

Primary murine calvarial osteoblasts (C.Ob) were isolated as previously described [24,25]. Briefly, calvaria were dissected from 3–5-day old C56BL/6J wild-type (WT) mice culled by cervical dislocation and cells isolated from the matrix by enzymatic digestion (αMEM (ThermoFisher, Loughborough, UK) containing 1 mg/mL collagenase d (Roche, Hertfordshire, UK)) for 30 min, followed by αMEM with 5 µM EDTA for 10 min and αMEM with 1 mg/mL collagenase d for 30 min. All steps were performed under sterile conditions at 37 °C. Cells were cultured under standard cell culture conditions (37 °C, 5% CO_2_) in basal osteoblast media: αMEM supplemented with 10% foetal bovine serum (Biosera, East Sussex, UK), 2 mM L-glutamine, 100 μg/mL streptomycin, and 100 U/mL penicillin (all from Sigma-Aldrich, Gillingham, UK).

### 4.2. Osteoblast Cell Lines

Murine osteoblast precursor stromal cell lines ST2 or MC3T3-E1 (CRL-2205 and CRL-2593 respectively, ATCC, Manassas, VA, USA) were cultured in basal osteoblast media (as described above) and used before passage 9. Human osteoblast cell lines hFOB 1.19 (CRL-3602, ATCC, Manassas, VA, USA) were cultured in in DMEM/F-12, no phenol red (Fisher scientific, cat: 21041025) supplemented with 10% FBS and 0.3 mg/mL G418 (Sigma-Aldrich, cat: 4727878001) at 35.5 °C for 3 days, before moving cells to 37 °C and culturing in osteogenic media (basal culture media plus 10^−8^ M menadione and 100 µg/mL ascorbic acid (all from Sigma-Aldrich) used before passage 20.

### 4.3. Isolation of Human Osteoblasts

Primary human osteoblasts were isolated from patients undergoing joint (knee/hip) replacement surgery due to osteoarthritis or fracture stabilisation surgery at the Royal Orthopaedic Hospital (Birmingham, UK: 16/SS/0172) or the University of Birmingham NHS Foundation Trust (ethical approval issued by the Human Biomaterials Resource Centre, Birmingham, UK: study number 21-376). Human osteoblasts (H.Ob) were isolated from all samples following protocols detailed by Davies et al.: bone chips (~2 mm^3^) were cut and cultured in DMEM supplemented with 2 mM β-glycerophosphate and 50 µg/mL L-ascorbic acid (all from Sigma-Aldrich) and 10% foetal bovine serum (Biosera) [26]. Cellular outgrowth occured between 10–14 days. Cells were used in experiments between passage (P) 1 and 4.

### 4.4. Induction of Osteoblast Mineralisation

MC3T3-E1, ST2, C.Ob, or H.Ob (8 × 10^3^ cells/well) were cultured for up to 21 days in osteogenic media [Basal media, 10 nM β-glycerol phosphate and 50 µg/mL L-ascorbic acid (all from Sigma-Aldrich)] with or without 1 µM sphingosine-1-phosphate (S1P). HFOB 1.19 was cultured for 8 days in osteogenic media (as described above), with or without 1 µM S1P (Cayman Chemicals, Ann Arbor, MI, USA). Eighty percent of the media was refreshed every 2–3 days. Mineralisation was assessed by quantifying alkaline phosphatase (ALP) activity: osteoblasts were lysed in RIPA buffer (Sigma-Aldrich) for 30 min on ice, harvested using a cell scraper, and centrifuged at 13,000× *g* for 10 min. Cell lysate was incubated with alkaline phosphatase yellow (pNPP) liquid substrate (Sigma-Aldrich) in a 1:4 ratio, in the dark with agitation for 45 min at 37 °C. ALP activity was quantified using a Synergy HT plate reader (Biotek, Winooski, VT, USA) with absorbance set at 405 nm. Data are expressed as percentage of control (%).

### 4.5. Culture of Human Mesenchymal Stem Cells (MSC)

Healthy donor primary human bone-derived mesenchymal stem cells (MSC) were purchased from Lonza Ltd. (Basel, Switzerland) at P2 and cultured in DMEM supplemented with 10% foetal bovine serum (Biosera).

### 4.6. Murine BMDM Isolation and Culture

Female 6–12-week-old C57BL/6 wildtype mice were purchased from Charles River and were maintained in a specific pathogen free facility, with free access to food and water. Environmental conditions were 21 ± 2 °C and 55 ± 10% relative humidity and a 12 h light–dark cycle. Hind limb tibiae from mice were dissected, and the bone marrow was collected though centrifugation at 10,000× *g* for 15 s in media containing RPMI, 1% FBS, and 1% Penicillin Streptomycin Solution (all from Sigma-Aldrich), as previously described [27]. Cells were cultured (1 × 10^6^) for up to 8 days in culture media (RPMI, supplemented with 10% foetal bovine serum (Biosera), 2 mM L-glutamine, 100 mg/mL streptomycin, 100 U/mL penicillin (all from Sigma-Aldrich)) for 72 h prior to treatment with M-CSF (50 ng/mL, Abcam, Cambridge, UK) and S1P (1 µM).

### 4.7. Human Monocytes Isolation and Culture

Blood was collected from healthy donors following written, informed consent and approval from the University of Birmingham Local Ethical Review Committee. The study was conducted in compliance with the Declaration of Helsinki. Human peripheral blood mononuclear cells (PBMCs) from healthy donors were isolated by two-step density gradient centrifugation as previously described [28]. Monocytes were negatively selected from PBMCs resuspended in MACs buffer using the EasySepTM Human Monocyte Isolation Kit (cat: 19359, Stem Cell, Cambridge, UK). Briefly, PBMCs (5 × 10^7^ cells/mL) were incubated with monocyte isolation cocktail (50 μL/mL) and platelet removal cocktail (50 μL/mL) for 5 min at room temperature. Magnetic beads (50 μL/mL) were added to the sample and incubated for 10 min before the samples were topped up to 5 mL using MACs buffer. Tubes were placed into the EasySep ‘The Big Easy’ magnet (cat: 18001, Stem Cell) for 5 min before all non-stuck cells (monocytes) were collected and 1 × 10^6^/mL were cultured overnight in human osteoclast differentiation media containing 50 ng/mL m-CSF (cat: 216-MC-025, R&D systems, Abingdon, UK). After 24 h, half the media was replaced with human osteoclast differentiation media containing m-CSF and RANKL (cat: 390-TN-010, R&D systems) at a final concentration of 25 ng/mL plus S1P (1 μM). Media was replaced every 3 days prior to assessment of osteoclast differentiation.

### 4.8. Osteoclast Staining

Cultured cells were fixed in 10% PFA for 15 min prior to incubation with TRAP staining solution (sodium acetate anhydrous, L-(+) tartaric acid, glacial acetic acid Napthol-AS-MX, 2-ethoxyethanol, and fast red violet LB salt) for 30 min at 37 °C. Wells were washed 3× in distilled water and then imaged using the Cytation 5 microscope (Agilent, Santa Clara, CA, USA) with Gen5 software (version 3.15). The osteoclast number-per-well was analysed using ImageJ (Version 1.54, National institute of health (NIH); Bethesda, MD, USA) and expressed as osteoclasts per 100 m^2^.

### 4.9. Osteoclast Resorption Assay

Osteoclasts were seeded (1 × 10^6^) into OsteoAssay plates (2B Scientific) in RPMI (Sigma-Aldrich) for 8 days with M-CSF (50 ng/mL, Abcam), with or without S1P (1 µM, Cayman Chemicals). Subsequently, they were removed by incubation in 10% H_2_O_2_ for 5 min. Plates were washed in PBS, allowed to air dry for at least 2 h, and then imaged using the Cytation 5 microscope (Agilent) with Gen5 software. Hydroxyapatite resorption was assessed by colour thresholding areas of resorption in ImageJ FIJI and calculating the percentage of resorption per area/well.

### 4.10. Protein Isolation and Western Blot

Protein isolation was performed using RIPA buffer (Sigma-Aldrich) containing 1× cOmplete™ Mini Protease Inhibitor Cocktail (Merck, Rahway, NJ, USA). RIPA buffer cocktail was added to cells for 30 min on ice under constant agitation. Debris was removed via a 12,000× *g* for 20 min and lysates were stored at −80 °C.

### 4.11. RNA Extraction and qPCR

RNA was extracted using the RNeasy Kit as per the manufacturer’s instructions (Qiagen, Manchester, UK). RNA 500 ng was converted to cDNA using a high-capacity cDNA reverse transcription kit following the manufacturer’s instructions (Applied Biosystems). RNA (500 ng) was converted to cDNA using a high-capacity cDNA reverse transcription kit (Applied Biosystems, Cheshire, UK) following the manufacturer’s instructions. Gene-specific Assay on Demand TaqMan FAM labelled primers (*β*_2_*M*–Mm00437762, *Acp5*–Mm00475698, *Ctsk*–Mm00484039, *Msr1*–Mm00446214, *S1pr1*–Mm00514644, *S1pr3*–Mm00515669, *S1pr4*–Mm00468695, *Sphk1*–Mm00448841, *Sphk2*–Mm00445021, *Spns2*–Mm01249328, *ACP5*–Hs00356261, CTSK–Hs00166156, *MSR1*–Hs00234007, *S1PR1*–Hs00173499, *S1PR3*–Hs01015603, *S1PR4*–Hs02330084, *SPHK1*–Hs01116530, *SPHK2*–Hs01016543, *IL6*–Hs00174131, *CXCL9*–Hs00171065, *CXCL10*–Hs00171042) were diluted 1:10 in a master mix (both Applied Biosystems), before plating into 384 LightCycler plates along with 2.75 µL of diluted (1:5) cDNA in diH_2_O. Samples were analysed in duplicate on the LightCycler 480 (Roche). Data were normalised to β_2_-microglobin and expressed as 2^−ΔCt^ or fold change (2^−ΔΔCt^) relative to untreated control.

### 4.12. RNA Sequencing

Following isolation of RNA, the RNA integrity number (RIN) was calculated using the High Sensitivity RNA ScreenTape^®^ (Agilent). Library prep was performed using the Lexogen QuantSeq 3′ mRNA-Seq Library Prep Kit FWD for Illumina (Illumina, San Diego, CA, USA). Sequencing was then performed using the NextSeq 500 (Illumina). Data quality was checked using FASTQC and MULTIQC packages in RStudio (Version 4.3.2). Trimming was performed using BBMap and FASTQC was performed to visualise the changes in RNA quality. A genome index was created using a preassembled genome from the ensemble. The “Mus_musculus.GRCm39.104.gtf” genome and corresponding primary assembly file were used to create the genome used for our analysis. Trimmed sequences were then aligned to the newly created genome using the STAR align package (version 2.7.5) and the Encode standard settings. MultiQC was performed on FastQC files generated from the aligned samples and feature counts were produced using the Subread package v2.0.1 to generate a table of gene counts, which could be analysed using DESeq2 v1.30.0 to explore differentially expressed genes. Feature counts were loaded into R and analysed using DESeq2. First counts per million (CPM) were calculated for each gene, and a threshold was generated to remove genes with a CPM value of <10. Contrasting genes with a *p* value of <0.05 were generated from the dataset and a histogram was produced to explore the range of *p* values across the sample. Differentially expressed genes were named using the ensemble database and analysed for Log_2_ fold change. Differentially expressed upregulated or downregulated genes were separated and pathway analysis was performed using gseGO or enrichGO from ClusterProfiler package 3.17.5 to identify key pathways, which were visualised using a range of plots.

### 4.13. Analysis of Publicly Available Sequencing Data Sets

Publicly available datasets GSE128423, GSE147287, GSE147174, and GSE162454 were analysed using the Seurat package (version 5.0.0) in R. Seurat objects were processed to remove cells with abnormal RNA counts and mitochondrial counts. Data were normalised and variable features were found between each cell. Samples were analysed using principal component analysis (PCA) and clusters were generated. Clusters were visualised using a TSNE plot and differences between the clusters were identified. The top differentially regulated genes from each cluster were used to manually identify the cell types in each cluster, which were renamed. Samples were then split into subsets to look at the cells of interest (GSE128423 and GSE147287—osteoblasts, GSE147174 and GSE162454—monocytes, macrophages, and osteoclasts). Gene expression changes in each subset were visualised using feature plots and dot plots.

### 4.14. Western Blot Analysis

Cells were incubated in RIPA buffer (Sigma-Aldrich) for 30 min on ice prior to storage at −80 °C. Total protein was assessed using a BCA assay (Thermo Fisher Scientific, Waltham, MA, USA) according to manufacturer’s guidelines. Samples were diluted 1:5 with SDS Laemmli buffer (Bio-Rad, cat: 1610737) heated at 95 °C for 10 min prior to loading into 10% gels (Nu-Page, cat: NP0326BOX). Proteins were separated using a 170 V current PowerPac Basic Power supply (cat: 1645050) for 50 min, then the gel was transferred on to a PVDF membrane using the Trans Blot Turbo mini 0.2 µm (cat: 1704156) and Trans-Blo turbo transfer system (all from Bio-Rad, Hercules, CA, USA).

Membranes were blocked with 5% milk (Marvel—Premier Foods, Tullamore, County Offaly, Ireland) diluted in PBS containing 1% TWEEN 20 (Sigma-Aldrich) for 1 h under agitation, then incubated with the following primary antibodies diluted in 5% milk for 1 h at room temperature: anti-Sphk1 (1:1000, Proteintech, cat: 10670-1-AP); anti-Sphk2 polyclonal (1:1000, Proteintech, cat: 17096-1-AP); and anti-B-actin (1:3000, Proteintech, cat: 66009-1-Ig). Membranes were then washed with PBS-T and incubated with the relevant HRP-conjugated antibodies for 1 h: mouse, 1:3000, cat: 1706516, (Bio-Rad), or rabbit (1:5000, cat: 401393, Merck Millipore). Membranes were washed as described and treated with Clarity Western Peroxide Reagent and Clarity Western Luminol/Enhancer Reagent 1:1 (cat: 1705061) prior to imaging using the ChemiDOC MP Imaging System (both from Bio-Rad). Band intensity was measured using ImageJ FIJI software.

## Figures and Tables

**Figure 1 ijms-25-05118-f001:**
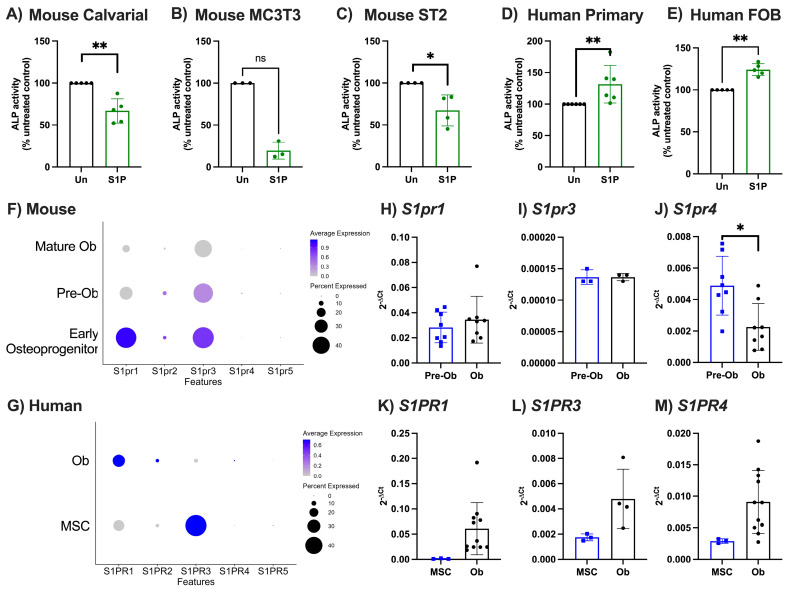
S1P increased osteoblastogenesis of primary human osteoblasts and decreased osteoblastogenesis of murine osteoblasts. (**A**) Murine primary calvarial (n = 5), (**B**) MC3T3 (n = 3), (**C**) ST2 (n = 4), (**D**) human primary isolated osteoblasts (n = 6), and (**E**) human fetal osteoblasts (n = 5) were either untreated (black) or treated with 1 µM S1P (green). (**A**–**D**) ALP activity was measured by ELISA and displayed as percentage of untreated control. (**F**) Clustered murine early osteoprogenitors, pre-osteoblasts (Pre-Ob), and mature osteoblasts (Mature Ob) were analysed for expression of S1P receptor genes. Dotplot of *S1pr1*, *S1pr2*, *S1pr3*, *S1pr4*, and *S1pr5* expression in each murine osteoblast subset, where circle size represents percentage of cells expressing the gene and colour indicates average expression value. (**G**) Clustered human MSCs and osteoblasts (Ob) were analysed for expression of S1P receptor genes. Dotplot of *S1PR1*, *S1PR3*, *S1PR4*, and *S1PR5* expression in each subset. (**H**–**J**) Calvarial osteoblasts were cultured in control (Pre-Ob, blue, n = 8) or osteogenic media (Ob, black, n = 8) for 8 days and gene expression of (**H**) *S1pr1*, (**I**) *S1pr3*, and (**J**) *S1pr4* normalised as 2^−ΔCt^ to *β*_2_*m*. (**K**–**M**) Human MSCs (blue, n = 3) and human primary osteoblasts (Ob, black, n = 11) gene expression of (**K**) *S1PR1*, (**L**) *S1PR3*, and (**M**) *S1PR4* normalised as 2^−ΔCt^ to *β*_2_*M*. Data are mean ± SEM, with each point representing an individual experiment/donor. * = *p* < 0.05; ** = *p* < 0.01 by Mann–Whitney (**A**–**E**) or paired *t*-test (**H**–**M**).

**Figure 2 ijms-25-05118-f002:**
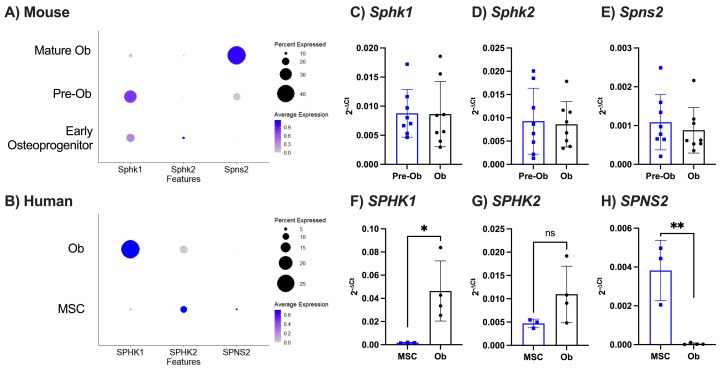
Murine and human osteoblasts express genes involved in S1P production. (**A**) Clustered murine early osteoprogenitors, pre-osteoblasts (Pre-Ob), and mature osteoblasts (Mature Ob) were analysed for expression of S1P production genes. Dotplot of *Sphk1*, *Sphk2*, and *Spns2* expression in each murine osteoblast subset, where circle size represents percentage of cells expressing the gene and colour indicates average expression value. (**B**) Clustered human MSCs and osteoblasts (Ob) were analysed for expression of S1P receptor genes. Dotplot of *SPHK1*, *SPHK2*, and *SPNS2* expression in each human osteoblast subset. (**C**–**E**) Calvarial osteoblasts were cultured in control (Pre-Ob, blue, n = 8) or osteogenic media (Ob, black, n = 8) for 8 days and gene expression for (**C**) *Sphk1*, (**D**) *Sphk2*, and (**E**) *Spns2* was normalised as 2^−ΔCt^ to *β*_2_*m*. (**F**–**H**) Human MSCs (blue, n = 3) and human primary osteoblasts (Ob, black, n = 11) were analysed for expression of (**F**) *SPHK1*, (**G**) *SPHK2*, and (**H**) *SPNS2* and data were normalised as 2^−ΔCt^ to *β*_2_*M*. Data are mean ± SEM, with each point representing an individual experiment/donor. (**C**–**H**) * = *p* < 0.05; ** = *p* < 0.01 by unpaired *t*-test.

**Figure 3 ijms-25-05118-f003:**
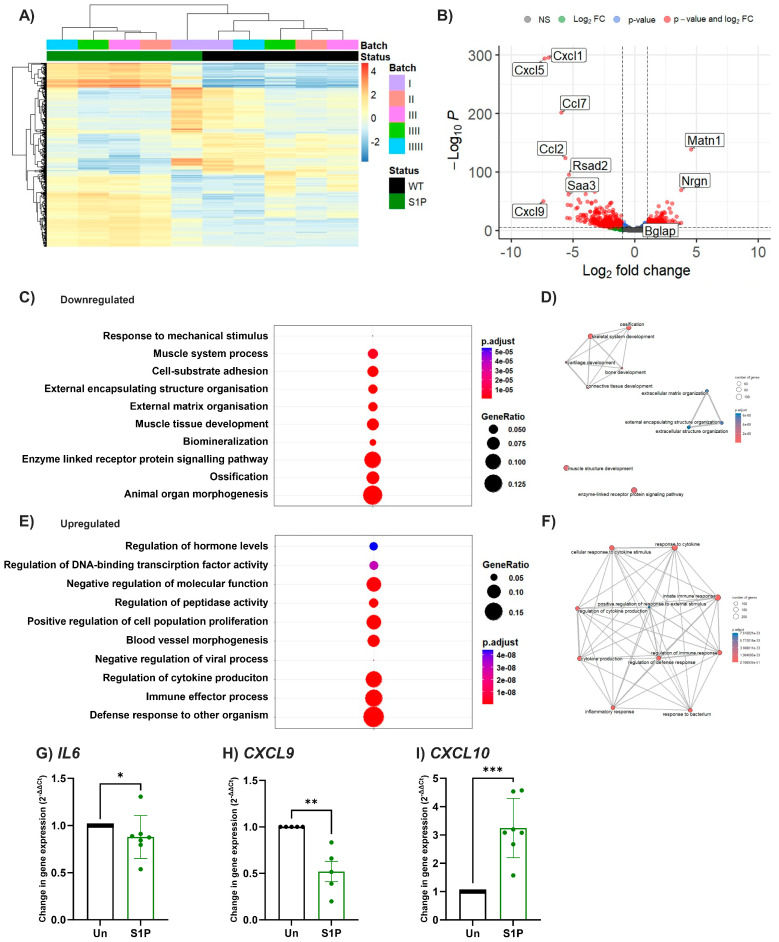
S1P upregulated inflammatory markers in murine osteoblasts. Bulk RNAseq analysis of primary calvarial osteoblasts following 8 days of differentiation in the presence or absence of S1P. (**A**) Heatmap and (**B**) volcano plot of differentially expressed genes following S1P treatment. Significantly different gene expression by both changes in *p*-value and log2FC values are shown in red. (**C**–**F**) Pathway analysis was performed on the use of clusterprofiler and gene set enrichment analysis. DotPlot of the top 10 (**C**) downregulated and (**E**) upregulated pathways ordered by adjusted *p* value. Circle size represents GeneRatio, whilst colour shows the adjusted *p* value. Emap plot of top 10 pathways (**D**) downregulated or (**F**) upregulated by S1P, with how the pathways map onto each other shown by connection lines, adjusted *p*-value by colour and number of genes involved by size of circle. (**G**–**I**) Human primary osteoblasts were differentiated and treated with or without S1P (1 µM) for 8 days. Gene expression of *IL6* (G, n = 7), *CXCL9* (H, n = 5), and *CXCL10* (I, n = 7) were normalised as 2^−ΔΔCt^ to untreated control. Data are mean ± SEM, with each point representing an individual experiment/donor. (**G**–**I**) * = *p* < 0.05; ** = *p* < 0.01, *** = *p* < 0.001 by Mann–Whitney U.

**Figure 4 ijms-25-05118-f004:**
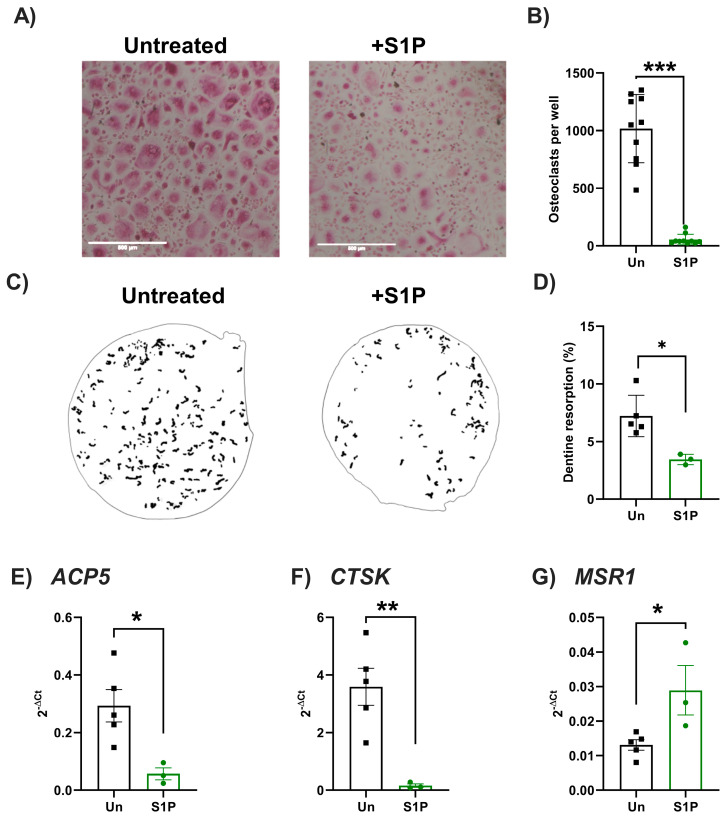
S1P reduced osteoclastogenesis of primary human monocytes. Human PBMC-isolated monocytes were treated with RANKL to induce osteoclastogenesis. (**A**,**B**) RANKL-treated cells were either (untreated, black bars, n = 10), further stimulated with S1P (green bars, n = 10). (**A**) Representative images of untreated or S1P-treated cells. (**B**) Osteoclast counts (TRAP-positive, multinucleated) in each condition and presented as number of osteoclasts per well. (**C**,**D**) RANKL-induced osteoclasts were cultured on dentine slices for 21 days in the presence of vehicle control (untreated, black, n = 6) or S1P (green, n = 3) prior to cell removal and staining of resorption pits. (**C**) Representative images of masks generated to show areas resorbed by osteoclasts. (**D**) The resorption area was recorded and displayed as a percentage of the total area. (**E**–**G**) Cells were untreated (grey, n = 5) or treated with RANKL only (black, n = 5) or treated with RANKL and S1P (green, n = 3 independent donors) for 8 days and RNA and gene expression of (**E**) *ACP5*, (**F**) *CTSK*, and (**G**) *MSR1* was normalised as 2^−ΔCt^ to *β*_2_*M*. Data are mean ± SEM from (B) n = 10 from 3 independent donors, (**D**–**G**) n = 3–5 independent donors. * = *p* < 0.05, ** = *p* < 0.01 and *** = *p* < 0.001 by paired *t*-test.

**Figure 5 ijms-25-05118-f005:**
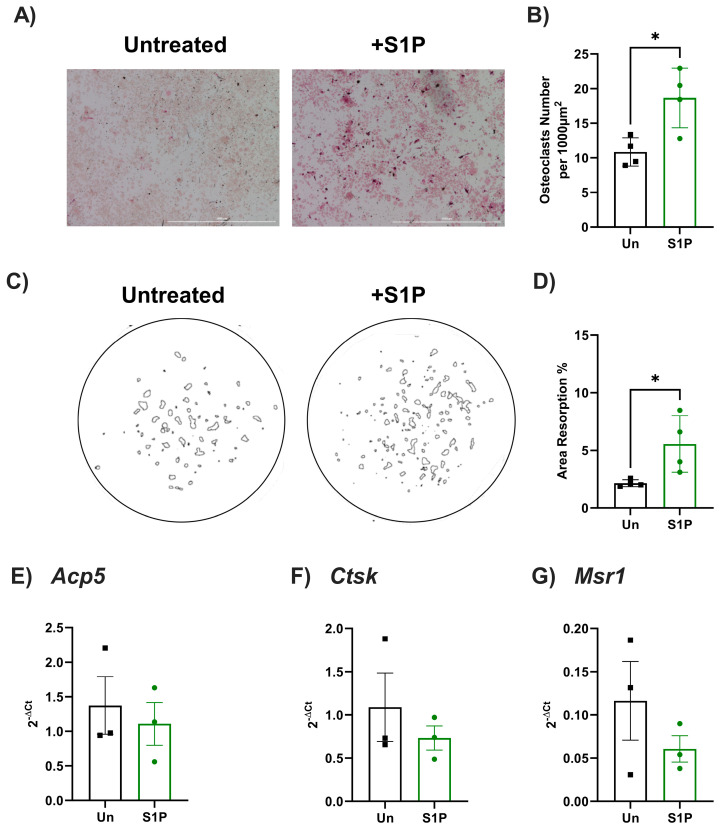
S1P increased osteoclastogenesis of primary murine monocytes. Murine bone marrow-derived monocytes (BMDM) were treated with M-CSF for 24 h before RANKL treatment. (**A**,**B**) RANKL-treated cells were either (untreated, black bars, n = 4), further stimulated with S1P (green bars, n = 4). (**A**) Representative images of untreated or S1P-treated wells stained for TRAP. (**B**) Osteoclast counts (TRAP-positive, multinucleated) in each condition and presented as number of osteoclasts per well. (**C**,**D**) RANKL-induced osteoclasts were cultured on hydroxyapatite coated plates for 8 days in the presence of vehicle control (untreated, black, n = 4) or S1P (green, n = 4) prior to cell removal and staining of resorption pits. (**C**) Representative images of masks generated to show areas resorbed by osteoclasts. (**D**) The resorption area was displayed as a percentage of the total area. (**E**,**G**) Cells were untreated (black, n = 5) or treated with RANKL only (black, n = 5) or treated with RANKL and S1P (green, n = 3 independent donors) for 8 days and gene expression for (**E**) *Acp5*, (**F**) *Ctsk*, and (**G**) *Msr1* were analysed and normalised as 2^−ΔCt^ to *β*_2_*m*. Data are mean ± SEM from (**B**) n = 10 from 3 independent donors, (**D**–**G**) n = 4 independent donors. * = *p* < 0.05 by unpaired *t*-test.

**Figure 6 ijms-25-05118-f006:**
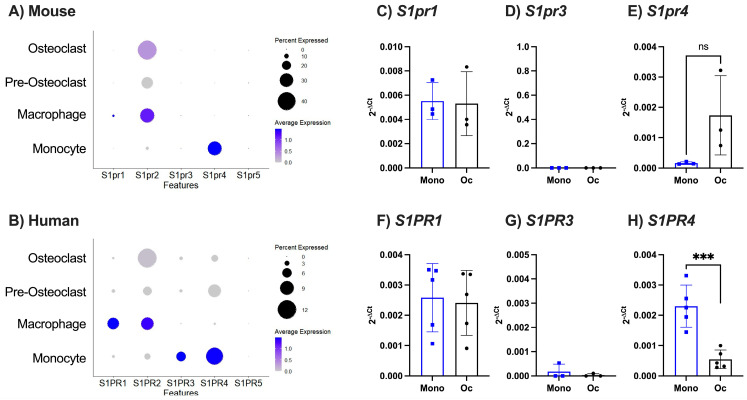
S1P receptor expression changes over osteoclastogenesis but not between species. (**A**,**B**) Analysis of publicly available single cell sequencing datasets (**A**) GSE147174 (mouse) and (**E**) GSE162454 (human) and looking at S1P receptors (S1PR1–S1PR5). Dotplot of clustered (**A**) murine monocytes, macrophages, pre-osteoclasts, and osteoclasts or (**B**) human monocytes, macrophages, pre-osteoclasts, and osteoclasts, showing expression of S1PR1-5, where circle size represents percentage of cells expressing the gene and colour indicates average expression value. (**C**–**E**) BMDMs cultured (8 days) with or without M-CSF and RANKL. Undifferentiated monocytes (Mono, blue, n = 3) or osteoclasts (Oc, black, n = 3), analysed for (**C**) *S1pr1*, (**D**) *S1pr3*, and (**E**) *S1pr4*. (**F**–**H**) Human monocytes derived from PBMCs cultured (8 days) with or without M-CSF and RANKL. Undifferentiated monocytes (Mono, blue, n = 5) or osteoclasts (Oc, black, n = 5), analysed for (**F**) *S1PR1*, (**G**) *S1PR3*, and (**H**) *S1PR4*. Data are displayed as 2^−ΔCt^, where ΔCt is the relative expression compared to *β*_2_*m* or *β*_2_*M* housekeeping gene. Data are mean ± SEM. *** = *p* < 0.001 by unpaired *t*-test.

**Figure 7 ijms-25-05118-f007:**
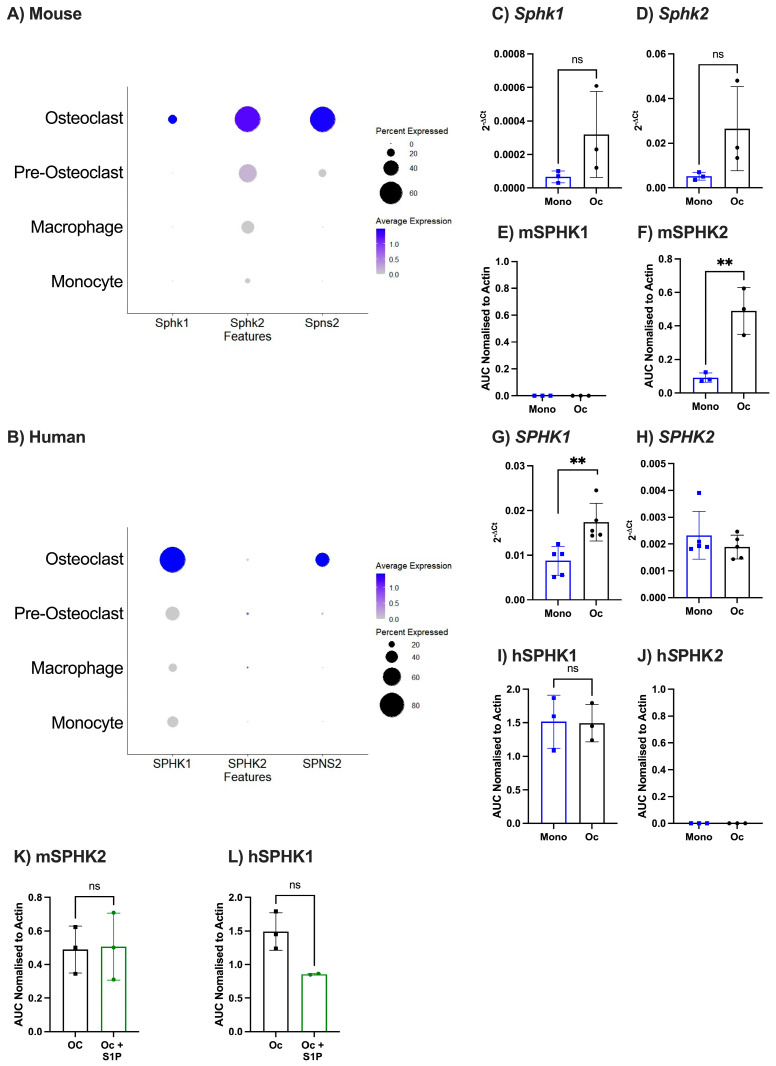
Kinases involved in S1P production vary over osteoclast differentiation and species. (**A**,**B**) Analysis of publicly available single cell sequencing datasets GSE147174 (mouse) and GSE162454 (Human) and looking at S1P producer (*Sphk1* and *Sphk2*) and transporter (Spns2). (**A**) Dotplot of clustered murine monocytes, macrophages, pre-osteoclasts, and osteoclasts, showing expression of *Sphk1*, *Sphk2*, and *Spns2*. (**B**) Dotplot of clustered human monocytes, macrophages, pre-osteoclasts, and osteoclasts, showing expression of *SPHK1*, *SPHK2*, and *SPNS2*. (**C**–**F**) Primary mouse bone marrow-derived monocytes cultured (8 days) with or without M-CSF and RANKL. Undifferentiated monocytes (mono, blue) or osteoclasts (Oc, black), analysed for (**C**) *Sphk1* (n = 3) and (**D**) *Sphk2* (n = 3) gene expression and (**E**) mSPHK1 and (**F**) mSPHK2 protein expression. (**G**–**J**) Human monocytes derived from PBMCs cultured (8 days) with or without M-CSF and RANKL. Undifferentiated monocytes (blue) or osteoclasts (black), analysed for (**G**) *SPHK1* (n = 5) and (**H**) *SPHK2* (n = 5) gene expression and (**I**) hSPHK1 and (**J**) hSPHK2 protein expression. (**K**,**L**) Primary mouse bone marrow-derived osteoclasts (**K**) or primary human osteoclasts (**L**) were treated with or without S1P and analysed for (**K**) mSPHK2 and (**L**) hSPHK1 protein expression. Gene expression data are displayed as 2^−ΔCt^ normalised to *β*_2_*m* or *β*_2_*M*. Protein expression data normalised to human or mouse β-actin. Data are mean ± SEM. ** = *p* < 0.01 by unpaired *t*-test.

## Data Availability

Data are contained within the article and Appendix A. The raw data presented in this study are available on request from the corresponding author.

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
