# Peer review of "The Species Effect: Differential Sphingosine-1-Phosphate Responses in the Bone in Human Versus Mouse"

_ijms, 2024, doi:10.3390/ijms25105118_

Round 1

Reviewer 1 Report

Comments and Suggestions for Authors

Comments to the Authors of manuscript number ijms-2916860 entitled “The species effect: Differential sphingosine-1-phosphate responses in the bone in human versus mouse.

This study highlights species-specific responses to S1P in osteoblasts and osteoclasts, with S1P favoring bone formation in human cells but promoting mineral resorption in murine cells.

1. L 33- small letter

2. L 35-37 – unclear mianly reviewed by.. -

3. L 50 – both references should be added

4. L 55- ICR ?

5. the hypothesis and goal of the study have to be clearly presented

6. L 70 - calvaria were dissected from 3–5-day old C56BL/6J wild-type (WT) 70 mice, but there is not added any information of ethical permission.

Honestly, I carefully reviewed the entire text up to this point. It's customary that when experiments involve animals, such as in this case where mice were euthanized at 3-5 days old, information regarding ethical approval for conducting such experiments is typically provided by the relevant ethical committee.

Properly obtaining approval from the ethics committee is crucial, especially when experiments are carried out on animals such as mice. Mice are not considered farm animals from which tissue collection during the slaughter process may sometimes bypass the need to obtain approval from the ethics committee, as is the case with animals intended for breeding purposes. In the case of neonatal, maternally dependent animals, such as 3-5 day old mice, killing is particularly delicate and requires special ethical consideration. Additionally, the lack of a specified method used for killing and collecting tissue raises doubts about compliance with ethical and humanitarian standards in the study. Therefore, to ensure compliance with ethical regulations and prevent animal suffering, obtaining approval from the ethics committee is essential.

Reviewer 2 Report

Comments and Suggestions for Authors

The study is interesting and well-executed. However, it lacks information regarding approval from the Local Ethics Committee for obtaining murine osteoblasts (C.Ob), BMDM, and human PBMCs. Although approval for human osteoblasts (H.Ob) was obtained from the appropriate committee, approvals for murine osteoblasts, BMDM and human PBMCs are absent. As a conscientious scientist, I cannot endorse conducting such procedures without proper ethical approval, nor can I support the publication of the results from such a study in a journal as reputable and esteemed within the scientific community as IJMS. Therefore, my recommendation at this stage of review is to reject this manuscript due to serious flaws.

Comments:

The references are neither cited nor formatted according to the journal's guidelines.

According to the journal's requirements, please provide detailed information about suppliers of all chemicals and reagents, including name, city, and location.

Ensure that all gene names are consistently italicized throughout the study.

L100, L102 At what temperature were the experiments conducted?

L150  Is the description of the RNA isolation procedure missing?

Fig 1B: the results of the t-test are missing.

L249-250: Add "sphk" (or "Sphk1 and Sphk2") and remove "(S1P)" since this abbreviation has been used already.

Fig 2C-H, correct the whiskers to make them more visible. Also, were no significant differences observed for SPHK2?

Fig 6C-E and 7C-D: the captions indicate that there were n=5 repetitions, but only n=3 data points are presented on graphs. The data on graphs 6E, 7C, and 7D suggest the existence of statistical significance, but no statistical significance was reported. Please verify.

Reviewer 3 Report

Comments and Suggestions for Authors

Review comments for ijms-2916860.

In this manuscript, the authors examined the system which relate to the difference in the effects of S1P on osteoclastogenesis between mice and human. The authors examined the difference of S1P-related systems, such as production, receptor, and inflammatory responses, between species. This manuscript would give some clue for clarification of species-related difference in the effects of S1P on osteoclastogenesis. However, the manuscript seems still developing stage and lack some key data.

The followings are comments.

Major

1. S1pr5

              Exploration for S1pr5 is also required, as peripheral blood leukocytes, the cells relate to osteoclastogenesis, express it.

2. No data for the change of inflammatory markers in human osteoblast.

              Though the authors showed the change of inflammatory markers by S1P in murine osteoblasts, there were no data for human osteoblasts. As the authors aimed to compare between mice and human, the data is mandatory to compare the inflammatory responses.

3. No comparison of the receptor expression in osteoclast/osteoclast precursors between species.

              Though the authors showed the expression of the receptor for S1P in osteoblast in figure 1, no comparison of the receptor expression in osteoclast/osteoclast precursors between species. Considering the experimental condition for figures 4 and 5, S1P would directly effect on osteoclast/osteoclast precursors. Therefore, the comparison of the receptor expression in osteoclast/osteoclast precursors between species are mandatory, though some data are expressed in figure 6.

              In addition, the authors exogenously added S1P in the cell culture for figures 4 and 5, and also the authors examined the system which relate S1P production in osteoclast/precursors, comparison before and after S1P stimulation is also necessary.

Minor

1. 3 cells were used for mice, but only primary cell for human. Is it possible to use some human osteoblastic cell line (osteosarcoma cells), such as Saos-2 or MG-63.

2. No protein level confirmation of the production between species.

Though the authors showed mRNA level expression of the enzymes that relate to the production of Sp1, no protein level confirmation was done. As post-transcriptional regulation could interfere with protein level expression, please confirm these expressions by protein level.

Also, please measure the amount of produced S1P in each cell, because final production is the coordination of these enzymes, and difference of the amount of produced S1P is more effective than the difference in the production enzyme.

Round 2

Reviewer 1 Report

Comments and Suggestions for Authors

i have not any comment

Reviewer 2 Report

Comments and Suggestions for Authors

Dear Authors,

Thank you for the corrections made, the manuscript is now much improved.

Best regards!